# Plasma proteome dynamics of COVID-19 severity learnt by a graph convolutional network of multi-scale topology

Samy Gauthier[1], Alexy Tran-Dinh[2,3], Ian Morilla[1,4]

**Efforts to understand the molecular mechanisms of COVID-19 have led to the identification of ACE2 as the main receptor for the SARS-CoV-2 spike protein on cell surfaces. However, there are still important questions about the role of other proteins in disease progression. To address these questions, we modelled the plasma proteome of 384 COVID-19 patients using protein level measurements taken at three different times and incorporating comprehensive clinical evaluation data collected 28 d after hospitalisation. Our analysis can accurately assess the severity of the illness using a metric based on WHO scores. By using topological vectorisation, we identified proteins that vary most in expression based on disease severity, and then utilised these findings to construct a graph convolutional network. This dynamic model allows us to learn the molecular interactions between these proteins, providing a tool to determine the severity of a COVID-19 infection at an early stage and identify potential pharmacological treatments by studying the dynamic interactions between the most relevant proteins.**

## Introduction

The sudden spread of severe acute respiratory syndrome coronavirus 2 (Sars-Cov-2) worldwide meant one of the most remarkable public health crises of recent times. Currently, more than 60 million individuals have been infected, causing over 1.5 million deaths related with severe complications of the COVID-19 disease.

There exists proved evidence on how certain proteins such as ACE2 receptor or TMPRSS2 are used by Sars-Cov-2 as entrance gates to infect the cell via membrane fusion and endocytosis (Ou et al, 2020; Zhou et al, 2020). Likewise, there are also multiple clues on a likely participation of other proteins in the downstream of the disease during its progression (Delgado Blanco et al, 2020; Yang et al, 2020; Zamorano Cuervo & Grandvaux, 2020; Scudellari, 2021). All these experimental efforts aim to characterise the progression of COVID-19 from an in situ baseline analysis of proteomic profiles. An approach that is getting more popular nowadays is to switch off those protein interactions resulting essential to the viral infection. Thus, the targeting of protein–protein interaction interfaces may be used to discover anti-COVID-19 treatment (Xiu et al, 2020; Yang et al, 2020). Unfortunately, this and most of those analyses tend to overlooking nonlinear programmes of interaction determined by subsets of proteins already described in such studies. Some of those programmes are merely contributing to an innocuous reconfiguration of the secondary immune response system, but others can be causally provoking a worsening in the severity of the symptoms (i.e., hyperinflammatory syndrome) during the disease progression.

In this work, we learn the latter through graph convolutional networks (GCNs) calibrated with higher topological features extracted from raw plasma proteomic data collected by the Massachusetts General Hospital Emergency Department COVID-19 (https://www.olink.com/mgh-covid-study/– [Filbin et al, 2021]). The examination of plasma as a potential source of insight into the evasion mechanisms of severe acute respiratory SARS-CoV-2 has been the subject of recent research (Cabrera-Garcia et al, 2022; Zhao et al, 2022). Plasma, as the liquid component of blood, is a complex mixture of substances, including antibodies and proteins produced by the host immune system in response to a stimulus, such as infection or disease. Alterations in the levels of these substances in the plasma may provide evidence of underlying conditions or disorders, even in the absence of direct evaluation of immune system cells.

Thus, we clustered and identified patient phenotypes by the World Health Organization (WHO)-mediated scores along their comorbidity effectors (if any) in a non-supervised fashion first and computed later with persistent homology of more than $1,400$ protein profiles in blood enhanced in endothelial cells (see Fig 1) across samples (Xu et al, 2019). To the best of our knowledge, this study is the first to chart longitudinal associations between plasma protein interactions and disease outcomes in patients with COVID-19 disease. Finally, we are convinced our results will be instrumental in a later experimental validation.

---

[1]Université Sorbonne Paris Nord, LAGA, CNRS, UMR 7539, Laboratoire d'excellence Inflamex, Villetaneuse, France   [2]Département d'anesthésie-Réanimation, INSERM, Université de Paris, AP-HP, Hôpital Bichat Claude Bernard, Paris, France   [3]Université de Paris, LVTS, Inserm U1148, Paris, France   [4]Department of Genetics, University of Malaga, MLiMO, Málaga, Spain

Correspondence: ian.morilla@uma.es; morilla@math.univ-paris13.fr

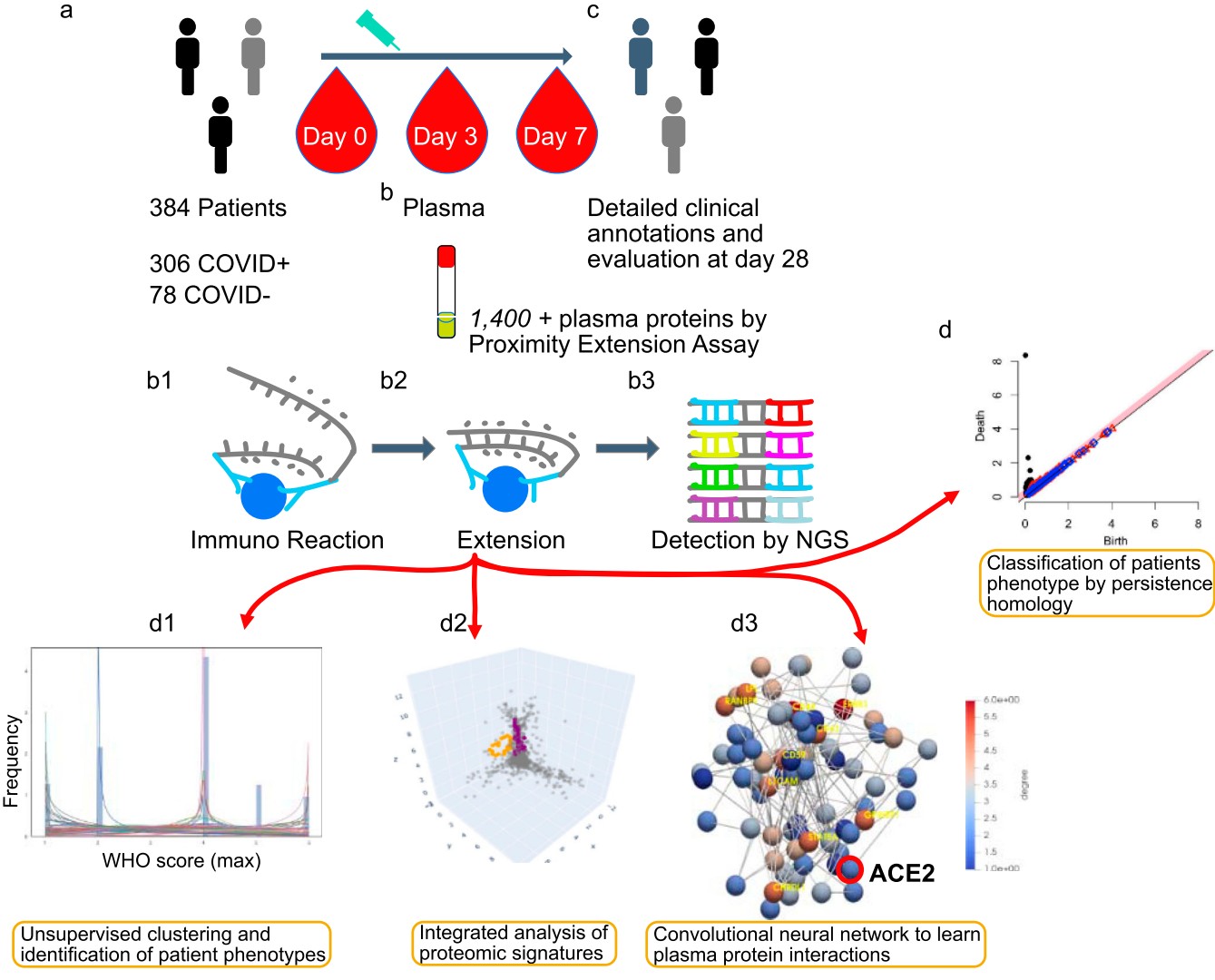

**Figure 1. Flowchart of the entire analysis.**
**(A)** Admission of inpatients presenting COVID symptoms and triage. **(B)** Blood extractions of inpatients at days 0, 3, and 7 after their admission. Measurement of 1,400+ plasma protein levels by proximity extension assay. (B1, B2, B3) Proximity extension assay protocol: immunoreaction, extension, and detection of levels by sequencing. **(C)** Record of clinical annotations and evaluation of discharged inpatients after 28 d of stay in the hospital. **(D)** The usage of algebraic invariants to study the shapes of each inpatient plasma proteome. (D1, D2, D3) Three stages of multi-topology analysis, namely, WHO scores of severity fitting and inpatient stratification based on that fit, protein candidates described by persistent homology per stratum, and prediction of disease progression based on convolutional neural networks constructed from those candidates and per stratum. Link https://www.olink.com/mgh-covid-study/. Modified and used with permission.

# Results

### Study design

Our models achieved tracking protein interactions occurring postinfection by which different levels of severity developed by 384 individuals suffering from COVID-19 symptoms might be explained (Filbin et al, 2021). In this sense, the COVID status of inpatients was tested positive prior to enrolment or during hospitalisation. Then, based on that test, we discriminated 306 patients as *Covid +* and 78 patients as *Covid −* (see Fig 2A). In addition, we wanted to improve our models' interpretability by means of the detailed clinical outcomes available from each

patient at day 28 of their stay in the hospital. In total, those variables encompass up to 40 different types of multi-variated sequences (see Supplemental Data 1). Those data allowed us to compute a precise overall score of severity based on discrete WHO scores (Organisation world health, 2021) provided in the cohort for each patient over time of stay in the hospital. First, we vectorised those scores with the aim of being using entropy (Gray, 2013) to calculate the information encapsulated by the WHO scores in each patient. That information in bits (Murphy, 2012) enabled the construction of a probability density function that basically well stratified individuals according to severity progression of the disease which ultimately reinforced the explanatory power of our learning omic models.

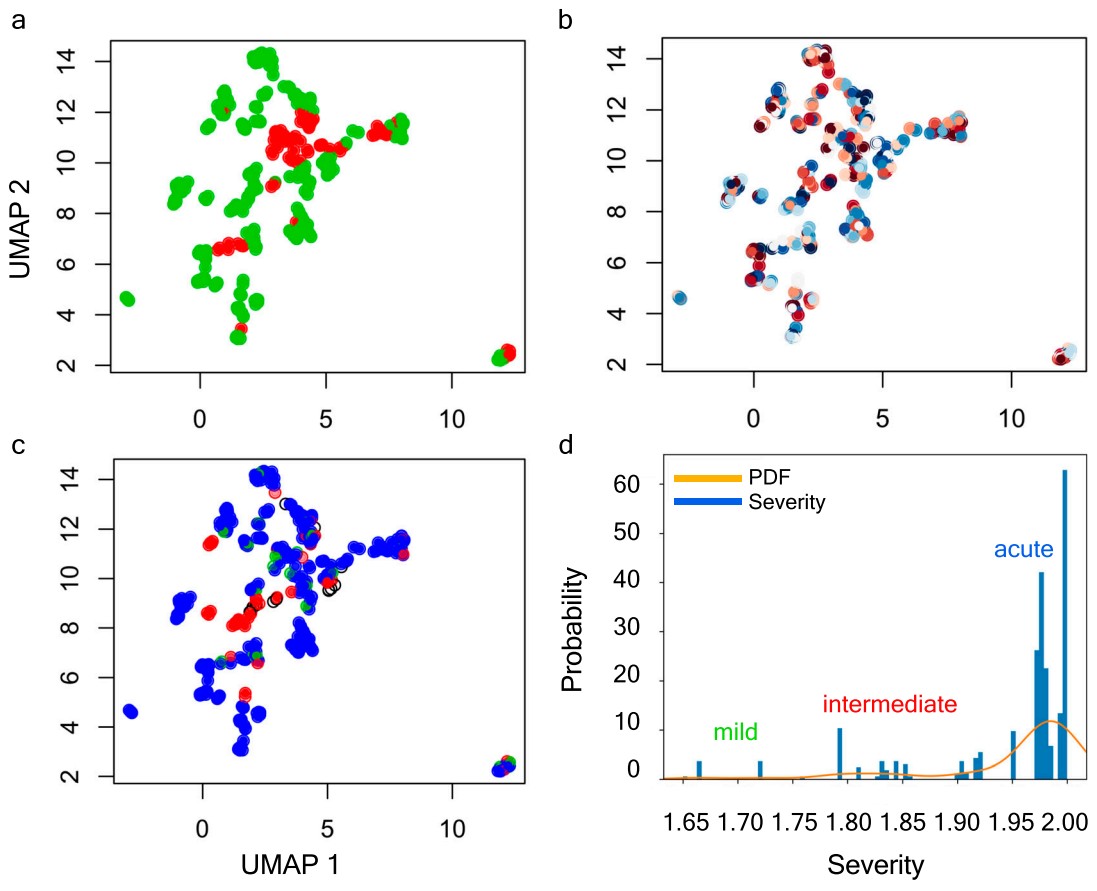

**Figure 2.  HDBSCAN severity clustering by means of entropy measures.**
**(A, B, C)** UMAP projection of clinical data. **(A)** Stratification by *Covid+* and *Covid–*. **(B)** Inpatients' stratification by WHO$_{max}$ score (i.e., a measurement correlated with the maximum WHO outcomes achieved by patients during their hospital stay and available in the clinical dataset). **(C)** Individual discrimination by Shannon's Entropy combined with *Hdbscan* clustering algorithm. The blank circles show inpatients considered as outliers by the dissimilarity of their symptoms. **(D)** Probability density function optimally fitted in accordance with Shannon's Entropy displaying three sharp peaks, namely, mild, intermediate, and acute associated with inpatients of the *Covid* cohort.

## Hierarchical models of WHO scale–based entropy information stratify patients by severity

WHO provides scores that monitor disease progression during a patient's stay at a hospital. These scores, recorded using the discrete measure $W_s = \{1, 2, ..., 6\} \in \mathbb{Z}^+$ at days 0, 3, 7, and 28 (see Table 2 and Supplemental Data 1), provide a local snapshot of an individual's disease progression (see Fig 2B). To better understand and track disease progression, we propose using information theory scores, as proposed by Shannon in his seminal paper (Shannon, 2001). This approach allows us to efficiently store and retrieve relevant information and create a continuous distribution to model disease progression. By vectorising the WHO scores and calculating entropy, we can extract self-information and fit the transformed data (see Fig S1A) to the best distribution by checking a comprehensive set of probability density functions. In particular, we obtain a discrete random variable $V$, with possible outcomes $v_1, ..., v_6$, which occur with probability $P(v_1), ..., P(v_6)$ and computed the entropy of each individual per vector by applying the formula $H(\hat{V}) = -\sum_{i=1}^{6} P(\hat{v}_i) log_{bits} P(\hat{v}_i)$ (see Fig 2C). The generalised continuous normal random variable yielded the best performance (see Fig

2D) and was used to calculate the bits of information (Mackay, 2017) for a context channel (i.e., $p(\hat{v}v')$) with $\hat{v}$ and $v'$ in the discrete alphabet $W_s$. Thus, we can accurately track disease progression and identify potential trends.

The generalised continuous normal random variable of subdomain $\mathcal{D} = [1, 2] \in \mathbb{R}^+$ was calculated as follows:

$$S_r(A, I)_i = \sum_{l=1}^{N} \frac{1}{D_{k,k}^{1/2}} A_{k,l} \frac{1}{D_{l,l}^{1/2}} I_l,$$

$$f(x, \beta) = \frac{\beta}{2\Gamma\left(\frac{1}{\beta}\right)} \exp - \|x\|^{\beta},$$

where $\beta \in [0.01, 0.99]$ and $\Gamma$ is the function gamma (Sun, 2020). By means of this function, we calibrated hierarchical clustering models that were computed by applying the Hdbscan algorithm (Campello et al, 2013). Hence, we achieved to discriminate the COVID cohort into three different groups (see Video 1, Video 2, and Video 3). Those groups basically met the *mild*, *intermediate*, and *acute* symptoms as registered in the available clinical dataset (see Fig 2D).

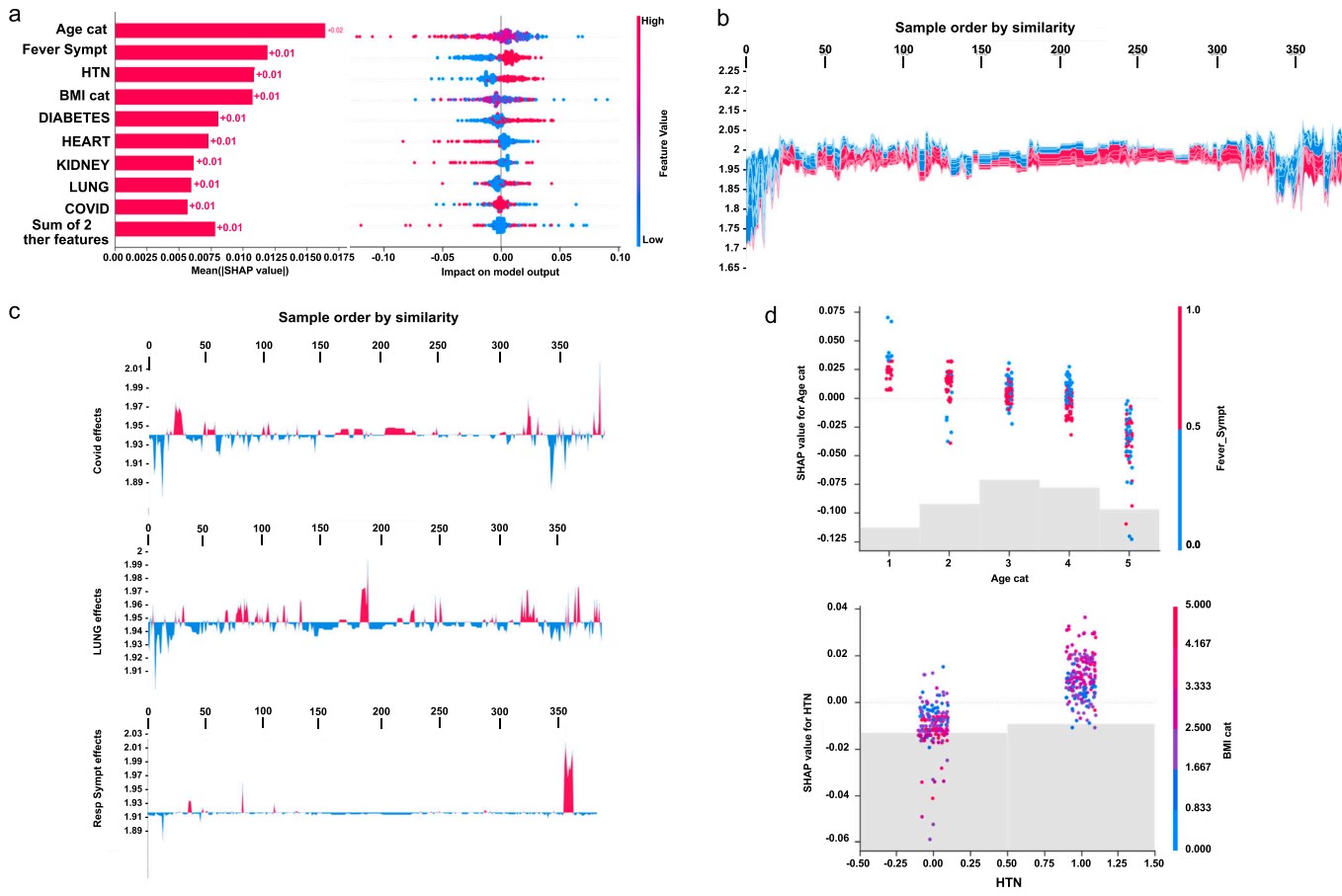

**Figure 3. Clinical evaluation of our entropy-based model on COVID severity score by the model and higher individual features.**
Initial explanations are based on a gradient boosted decision tree model trained on the COVID cohort. **(A)** Left: bar chart of the average *SHAP* value magnitude. *Age* was the most influential symptom, changing the predicted absolute COVID probability on average by two percentage points (0.02 on *x*-axis). Right: a set of beeswarm plots, where each dot corresponds to an inpatient in the cohort per significative symptom. The dot's position on the *x*-axis shows the impact that a symptom has on the model's prediction for a given inpatient. The piled-up dots mean the density of inpatients suffering from a symptom with similar impact on the model. Younger ages reduce the predicted Covid risk, elder ages increase the risk. **(B)** Globally stacked *SHAP* explanations clustered by explanation similarity. Inpatient profiles land on the *x*-axis. Red values increase the model prediction, blue ones decrease it. Two clusters stand out: On the left is a group with low predicted risk of suffering an acute Covid, whereas on the right, we have a group with a high predicted risk of suffering from acute COVID. **(C)** Top-bottom: locally stacked explanations clustered by explanation similarity for infection, lung, and respiratory symptoms. **(D)** Effect of a single feature across the whole cohort. Top–bottom: dependence plots for *Age* and *Hypertension* (HTN) features. These plots display inflection points in predicted age and hypertension as $Age\ cat$ and *HTN* (oldness by years on average and hypertension complaint per individual in the cohort) changes. Vertical dispersion at a single category of *Age* (resp. *HTN*) represents interaction effects with other features. To help reveal these interactions, we coloured by *Fever* (resp. *BMI*). We passed the whole explanation tensor to the colour argument in the dependence plots to pick the best feature to colour by. In this case, it selected fever symptoms (resp. Body Mass Index) because that highlights that the average age (hypertension) per inpatient has more (less) impact on Covid severity for categories with a low (high) *Fever* (*BMI*) value.

From this stratification, 13 out of 384 patients were excluded to be considered as outliers with noisy data. These patients displayed dissimilar symptoms and unmatched characteristics amongst them to be included in any of the groups (see Video 4 and Fig S2).

## The latent space of clinical features explains patients' stratification

We unified many local perspectives of the clinical dataset to explain models of severity progression (see Fig S3A–D). To this end, we summarised both an entire model and individual features learnt from the pdf of our Shannon's entropy severity. This task was eventually performed using the medical outcome dataset (see Supplemental Data 2) to train a three-dense layer convolutional

neural network with 269,313 trainable parameters (see Fig S4). The architecture of this network consisted of two convolutions *2D* and a last flatten layer with a scheme of (384, 40, 512) × 2 and (384*(40//512)*1) as output dimension. Next, we computed local explanations based on Shapley-related extensions, that is, the so-called SHAP values (Lundberg & Lee, 2017). To figure out the relative contribution of each feature to our model output individually, we plotted the values of every local feature for every sample in the cohort. The Fig 3A—right hand panel—shows a plot of sorted features by the sum of local value magnitudes over all samples and uses such values to show the distribution of the impacts each feature has on the model output. The colour represents the feature value (red high, blue low). This reveals, for example, a known fact, that a high *categorical age* (% lower status of the population) raises

the predicted risk of experiencing acute severity in the COVID disease. In the left-hand panel of Fig 3A, we observe this same effect in stacked red bars for our multi-class output task.

To understand how a single feature affects the output of the model, we plotted the local value of that feature versus the value of the feature for all the inpatients in the clinical dataset (see Fig 3B). Now, we can zoom in some of those effects individually as shown in Fig 3C for the lung and respiratory (the one quantified as lowest in its contribution to severity learning) symptoms. Because locally explained values represent a feature's responsibility for a change in the model output, the plot in Fig 3D represents the change in predicted COVID severity as *Age cat* (the average age per category in the cohort), or preexisting hypertension change. Vertical dispersion at a single value of *Age cat* represents interaction effects with other features. To help reveal these interactions, we can colour by another feature. If we pass the whole explanation tensor to the color argument, the scatter plot will pick the best feature to colour by. In this case, it picks *Fever_Sympt* (symptoms associated with fever) because that highlights that the the average age per category in the cohort has less impact on acute COVID severity for categories with a high *Fever_Sympt* value.

The values of interaction between locally explained variables are a generalisation of those to higher order interactions. Fast exact computation of pairwise interactions is implemented for tree models. This returns a matrix for every prediction, where the main effects are on the diagonal and the interaction effects are off-diagonal. These values often reveal interesting hidden relationships, such as how the increased risk of death peaks for inpatients with mild febrile symptoms at the age between 20 and 34 years (see Fig 3D -upper panel-), or that non-preexisting hypertension has less impact on individuals with a high *BMI_cat* value (see Fig 3D -lower panel-).

## Persistent homology identifies novel key proteomic features involved in severity

Based on the previous clinical characterisation of COVID severity, we exploited the proteomic plasma information available for the remaining 371 individuals in the cohort. Unfortunately, the particular geometry of inpatient's proteomes as embedded onto lower dimensional spaces resulted highly sensitive to parameter setups considered in downstream analyses according to entropy-based severity. In such a scenario, we computed topological invariant structures instead (see Video 5). These invariants, the so-called simplicial complex, qualitatively analyse features that persist across multiple scales. Such invariants can be classified over days 0, 3, and 7 by obtaining their generators through persistent homology (see Fig S5A). This analysis led us to identify unique protein configurations (see Figs S5B and S6) within inpatient proteomes based on their connected components (Xia & Wei, 2014; Aktas et al, 2019). Thus, the whole universe of proteome embeddings could be enclosed in the quotient space $\mathcal{P}/\sim$ under the given equivalent relation: for any $p.\in\mathcal{P}$, $p_i\sim p_j$ if the projected proteome $i$ "is similar to" on the set of all rotated tails, the so-called special orthogonal group $SO(3)$ (Hall, 2004). Hence, a partition with two classes of equivalence come out whose disjoint union determines the all three revealed groups of patients (see Video 6). Indeed, these classes of equivalence have as represents $[c_b]: = \{x\in\mathcal{P} : x\sim c_b\}$ if $b$ is a ball-like shape in $SO(3)$ and $[c_s]: = \{x\in\mathcal{P} : x\sim c_s\}$ if $s$ is a start-like shape in $SO(3)$. These two classes can be visualised in upper and lower panels of Fig 4A and B. Now, to identify proteins whose profiles are invariants of each inpatient over the days of their stay $\{0, 3, \text{and } 7\}$, we primary analysed the classes of equivalence $[c_b]$ and $[c_s]$ of each partition by persistent homology per group. Specifically, we used persistence diagrams wherein we quantified the number of homology generators (see upper Fig 4C) while testing their quality by means of confidence band generated by probabilistic boosting and density diffusion (see lower panel of Fig 4C). This enabled the identification of a unique set of proteins (see Video 1, Video 2, and Video 3) that encapsulated two- and three-dimensional structures (Fig 4D and E) important to topologically characterise severity classification per group over the days of stay of each individual in the hospital. Thus, we mapped the transmembrane serine proteases TMPRSS5 and SS15 (Huttlin et al, 2017) as novel receptors taking part of the infection machinery. These proteins belong to the same family of TMPRSS2, a known receptor used by Sars-CoV-2, to enter the cell (Hoffmann et al, 2020). Both of those proteins were located amongst the dysregulated interactome of inpatients stratified as acute and strikingly also as mild. Furthermore, amid the proteins we found (see Video 1, Video 2, and Video 3 for the entire list), there were proteins in acute -BRK1, LAP3, SLC27A4, SLC39A14-, intermediate -SLC27A4-, and mild -BRK1, SLC27A4, and SLC39A14- inpatients functionally linked (Huang et al, 2009a, 2009b) with AP3B1, BRD4, BRD2, CWC27, SLC44A2, and ZC3H18 whose profiles are reported to be likely involved in early infection caused by the virus (Gordon et al, 2020). Finally, the functional analysis of the novel proteomic features resulted from our persistence analysis showed along with their ancestors two sharp clusters bound to pneumonia and inflammation pathways (see Fig S7).

## Dynamic tracking of protein interactions required by the virus to efficiently infect the cell

Once we put the spotlight on individual proteins topologically important to discriminate COVID patients over time, we envisaged to capture their dynamics of functional interactions at regulatory levels. To this end, integrated protein–protein interaction networks were firstly constructed per group of patients using colocalization, coexpression, physical interactions, and shared domains (Morilla et al, 2010; Warde-Farley et al, 2010). Then, we enquired these graphs about their connection quality by means of degree and centrality distributions as shown in Fig 5. Surprisingly, we could confirm (see Fig 5) that it neither was highly connected nor played an important modular role in the graphs. Next, to predict disease progression, we monitored the behavioural regulation of the nodes' graph aggregation on semi-supervised learning on a community composed by known and unknown protein interactions with ACE2 and TMPRSS2 (Morilla et al, 2022) (see the Video 1, Video 2, and Video 3, and Video 4). To compute such a tracking, we endowed the graphs with a tailored hybrid design (see Fig S8) covered in convolutional layers along with a spectral rule (Defferrard et al, 2016) as occurs in GCNs. The first model's performance yielded an accuracy, for each group of inpatients, of 0.49, 0.41, and 0.85 supported by 76, 146, and

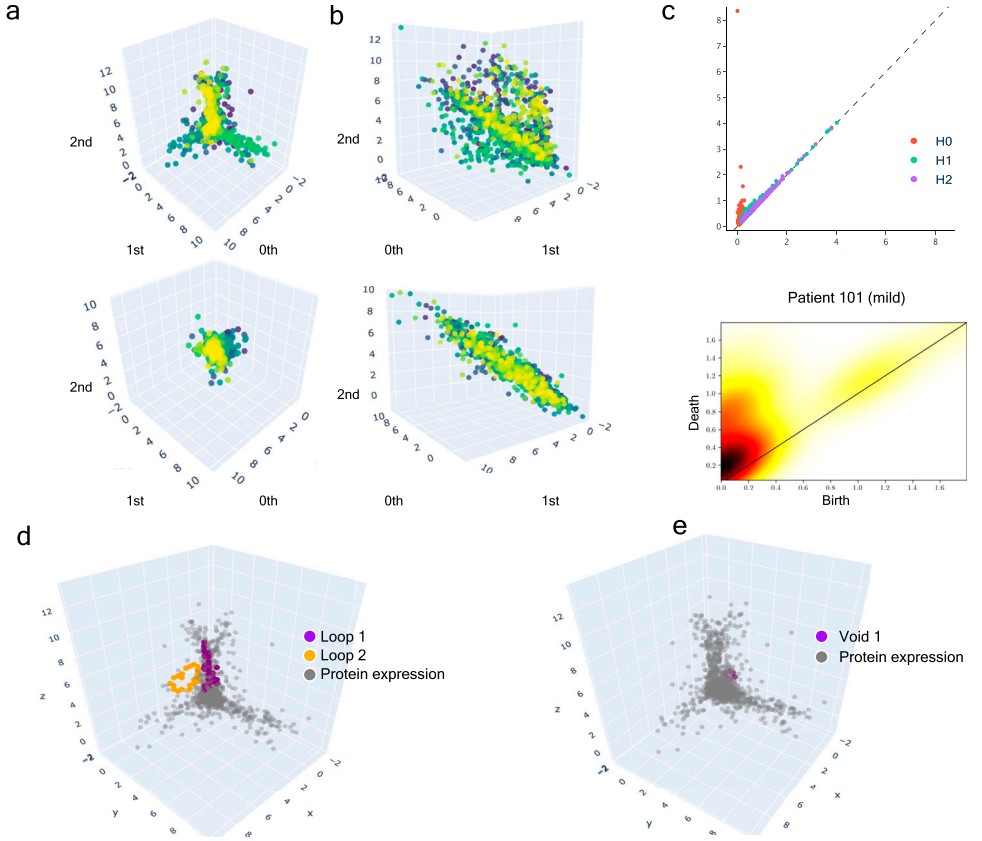

**Figure 4. Multi-scale topology analysis flowchart.**
**(A)** Upper: class of equivalence $[c_b]$ determined by umap projection of mild inpatients upon rotation on *SO(3)*. Lower: umap projection of an inpatient's soluble proteome. **(B)** Upper: class of equivalence $[c_t]$, taking as example to show the mild inpatient 101. Lower: umap projection of that inpatient's soluble proteome. **(C)** Upper: topological feature extraction from diagram of persistence of patient 101 and its later calibration. Lower: application of density diffusion for separating noise from robust signals in the persistence diagram of impatient 101. **(D)** Spotted loops of proteins enclosing dimension 2 structures important to explain severity stratification over time of patient 101. **(E)** Spotted voids of proteins enclosing dimension 3 structures important to explain severity stratification over time of patient 101.

355 samples regarding *covid* and *non–covid* feature representations, respectively (see Figs S9–S11 and Table 1). Those values increased to 0.71, 0.84, and 0.95, respectively, to the second learning model. We also found that their corresponding performances asymptotically tended to 0.65, 0.7, and 0.81 when the layers were largely increased from the 32 units in the convolutional architecture.

In that way, we learnt how ACE2 and TMPRSS2 interacted with the persistent novel candidates to explain the virus machinery at its entrance into the cell to put patients into mild, intermediate, or acute groups of severity over time (see Video 7, Video 8, Video 9, and Video 10). Hence, a primary set of proteins that led to acute severity consisted of the progressive aggregation of BCAN, CA2, CA12, CLEC4, FOLR1, FOLR2, IFNGR2, IGSF3 (R), ILR13A1(R), LAIR1, LRRN1, PCDH17, RTBDN, SEZ6L, SIGLEC6 (Schulte-Schrepping et al, 2020), and TNFRSF21 with respect to *covid* feature representation. Especially, CLEC4 belongs to a protein family (i.e., the C-type lectin receptor) involved in regulating immune reactivity through platelet degranulation whose expression significantly decreased in COVID-19 and correlated with disease severity (Overmyer et al, 2021). The interactions occurring early on during the infection amongst AXL, CD58, DDR1, DLK1, FCGR3A, TNFRSF12A, UXS1, and XPNPEP2 set the *non–covid* latent feature. Herein, we spotted CD58, a nonclassical monocyte, such as CD274 (PD-L1) known inhibitor of T-cell activation along with Arginase 1 (Bronte et al, 2003; Li et al, 2018) highly expressed in neutrophils in COVID-19 patients or CD24 involved in neutrophil degranulation with an increased expression of neutrophil function (Overmyer et al, 2021). Therein, we also identified FCGR3A (encoding CD16a) that is regulating severity-dependent alterations of the myeloid cell compartment during Sars-CoV-2 infection. Indeed, FCGR3A has been already found to be a nonclassical monocyte marker in COVID-19 (Schulte-Schrepping et al, 2020). Next, to the *covid* representations that determined intermediate severity of patients, we found the early aggregations of FR2, GALS4, IL1RN(R), ILR1(R), LRPAP1, RNF41, TRIM21(R), and VWV2. Remarkably, RNF41 plays a central role during interactions of Sars-CoV-2 with innate immune pathways because its interferon pathway is targeted by RNF41 (NSP15) (Gordon et al, 2020). Then, the intermediate *non–covid* representations are governed by the interactions of CCR5 (intestinal pro-inflammatory), CPA1, LAMA2, PLA2G4A, PON3, SETMAR, TGFB1, and XCL1. Finally, aggregations of C4BPB, CD70(R), IPCEF1, MAVS(R), PLCG2, and THBS2(R) led to mild severity stratification to *covid* feature representations. At the same time, the interactions between CD200, MAPKAPK5 (Kindrachuk et al, 2015), NTRK3(R), PRAP1(R), and XPNPEP2(R) set the *non–covid* feature representations. In these two lists, we might mention a similar effect on neutrophils and expression as CD58 to CD70 and CD200 (Overmyer et al, 2021).

We checked that *covid* feature representation of patients with acute symptoms was functionally charaterised by a set of proteins involved in the fusion of virus enclosure to the host endosome membrane (GO:0039654) at the virus entrance into the cell. Overall, the interactions between ACE2 and TMPRSS2 and these persistent proteins were largely enriched in the immunoglobulin-like fold

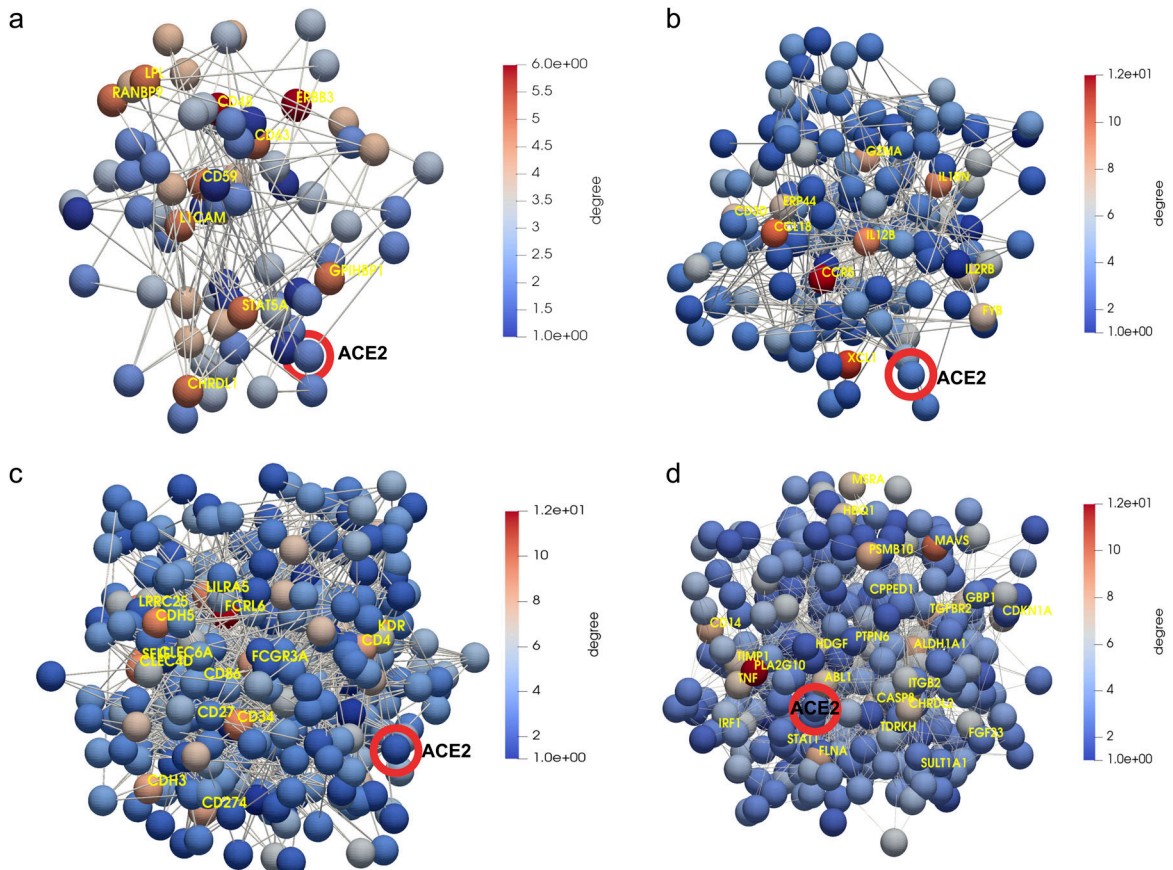

**Figure 5. Regulatory gene networks predicting disease progression regarding ACE2 and TMPRSS2.**
**(A)** Mild patients. **(B)** Intermediate patients. **(C)** Acute patients. **(D)** Group of proteins nonfunctionally enriched in acute patients. Highlighted in red ACE2 as one initial seem in the downstream analysis of protein interactions occurred post-infection. Yellow enhances those proteins (ERBB3, CD48, CCR5, FCRL6, and PLA2G10, among others) with a higher connectivity degree in the networks.

functional category. From a mere clinical stratification point of view, these conditions characterise most of the inpatients in the available clinical dataset suffering of cardiovascular complications. *Non–covid* representation of acute patients was strongly composed by membrane and signal peptide functional categories, protein tyrosine kinase and glycoprotein with extracellular and cytoplasmatic topological domains and transmembrane helix and integral component in the region biological processes. In this case, these conditions were felt on patients mainly suffering from diabetes of type 1 and immunosuppression.

Regarding the *covid* feature representation of intermediate patients, an overabundance of protein binding function is observed with diabetes of type 2 and normal variation diseases associated with such conditions. On the other side, the *non–covid* feature of this group is strongly enriched with disulfide bond and signal peptide functional categories. Therein, we found various terms directly linked with endosome viruses' machinery. Thus, we identified clathrin-dependent endocytosis (GO:0075512), host lysis, inhibition of host IKBKE, JAK1, RLR pathway, TBK1, and TLR pathway triggered by the virus in the host cell. Those symptoms were identified to a wide range of the complications described in the clinical dataset; in particular, asthma severity, chronic hepatitis C,

**Table 1. Performance of the GCN hybrid model to mild COVID patients.**

|  | Precision | Recall | f1-score | Support |
|---|---|---|---|---|
| False | 0.07/0.33 | 0.17/0.06 | 0.10/0.10 | 18 |
| True | 0.95/0.95 | 0.89/0.99 | 0.92/0.97 | 337 |
| Accuracy |  |  | 0.85/0.95 | 355 |
| Macro avg. | 0.51/0.64 | 0.53/0.52 | 0.51/0.53 | 355 |
| Weighted avg. | 0.91/0.92 | 0.85/0.95 | 0.88/0.93 | 355 |

The first set of figures before the "/" represents Model 1, whereas the second set of figures represents Model 2 with aggregation (see in Materials and Methods section: notes on the GCN). For the sake of simplicity, we only show acute subgroup patients. For further details, see Tables S1 and S2.

immunosuppression after liver transplantation, diabetes, especially strong of type 2, heart and kidney complications, and hypertension.

Finally, the latent *covid* representation of mild patients' proteomes was functionally characterised by a weak overabundance of disulfide bond and glycosylation site (i.e., N-linked as GlcNAc, etc.). These categories were related with suppression by the virus of host-adaptive immune response (GO:0039504). Remarkably,

there were no disease-associated genes type-specific to these biological processes. The *non–covid* mild features were actually overrepresented by signal peptide, qualitatively similar to those features described to the intermediate *non–covid* patients. We will fully expose and discuss the intriguing implications of such results in the next section.

## Discussion

Sars-CoV-2 has become in these two last years a real life-thread that has collapsed the health systems worldwide. Many efforts have been already done to structurally characterise the Sars-CoV-2 spike protein. To predict its severity, large mappings of proteins likely involved in the machinery applied by the virus to infect the cell have been reported (Jackson et al, 2022; Sokhansanj & Rosen, 2022). All these investigations have led to enormous advancements in COVID-19 treatment that consequently have given rise to efficient vaccines (Kumari et al, 2022). However, there is still some facets not well-characterised or yet sufficiently explored. In our attempt to contribute to this research, we computed an overall severity score based on WHO scales instrumental to provide a chart explaining the protein interactions required by the virus to stratify a patient's infection into mild, intermediate, or acute (Organisation world health, 2021). To this end, we made use of a double analysis linking symptoms to protein expressions and interactions with ACE2 and TMPRSS2. Merely from a stratification standpoint, we give novel information and verified known facts about COVID. As conclusion, we could claim that in itself, COVID-19 is not as harmful as it is in association with other risk factors such as age, febrile symptoms, or overweight. Indeed, most of the *Covid–* patients, though some were considered as outliers as indicated in the earlier sections, held a relative high entropy value of severity due to an eventual intubation, ventilation or supplementary oxygen requirement. To achieve those high peaks, *Covid +* should be overweight elderly people and present fever and/or respiratory symptoms.

More importantly, we obtained functional evidence of how particular sequences of proteins interacted with the virus to block the immune systems during the infection. Thus, we could explain the infection fate towards the acute symptoms because an endosome acidification is produced during the infection initiating conformational proteins fusion (Yang et al, 2022). In that way, Sars-CoV-2 could take advantage of such pathway to be endocytosed as happens with many types of viruses such as influenza A virus, alphaviruses, or HIV-1.

Regarding patients suffering from intermediate symptoms that endocytic process is caused over proteins that contain at least one coiled domain forming stiff bundles of fibres. Hence, proteins are modified by signaling upon creation of interchain disulfide bonds, which can produce stable, covalently linked protein complexes likely contributing to fold and stabilise proteins. In virus internalization, clathrin-mediated endocytosis could be then generated in response to getting assembled on the inside face of the cell membrane to cleave the host cell (CCV) by the action of DNM1/dynamin-1 or DNM2/dynamin-2 (Jima & Hinshaw, 2018). Then, the virus may be delivering their content to early endosomes via CCV.

These mechanisms could be expressing in Sars-CoV-2 using different ways as by lysing the host cell, blocking the host innate defenses via IKBKE/IKK-epsilon kinase inhibition, JAK1 protein, DDX58/RIG-I-like repector (RLR) what stabilises the antiviral state, TBK1 kinase inhibition to prevent IRF activation, or toll-like recognition receptor (TLR) pathway evasion, which makes the production of interferons to be inhibited and so to establish a stable antiviral state (Chen et al, 2021; Aliyari et al, 2022).

To mild cases, the Sars-CoV-2 protein could be preventing the tuned repertoire of self and nonself antigens' recognition of efficiently acting against the malicious effects of cell infection. In these cases, Sars-CoV-2 would be escaping the adaptive immune response by simple interference with the presentation of antigenic peptides at the surface of infected cells (Sette et al, 2021).

The use of plasma in COVID-19 research can be justified for many reasons, even if the cells of the immune system are not evaluated directly. Plasma contains a wide range of substances produced by the immune system, including antibodies and other proteins, which can provide valuable information about the body's response to the SARS-CoV-2 virus. By studying these substances in the plasma, researchers can gain insights into the immune system's response to the virus and how it may help or hinder the body's efforts to fight off the infection. Additionally, plasma can be collected and stored easily, making it a convenient and accessible source of samples for researchers studying COVID-19. Finally, plasma can be used in a wide range of experimental techniques, making it a versatile tool for researchers studying the immune system. However, it is important to note that the relationship between plasma findings and the immune system's response is not always clear, and further research may be needed to fully understand this relationship.

The approach presented in this work provides a more comprehensive and dynamic understanding of the molecular mechanisms of COVID-19 compared to more conventional methods such as antibody testing and PCR testing. These classic methods are limited in their ability to assess disease severity and provide a snapshot of the disease state, whereas the use of protein level measurements and clinical evaluation data, combined with GCNs, provides a dynamic understanding of the molecular interactions between proteins and can help determine the severity of a COVID-19 infection at an early stage.

Whereas traditional approaches rely on identifying the presence of the virus through antibodies or PCR, this study goes beyond that to provide insights into the molecular interactions and changes in protein expression that occur during the progression of the disease. This can aid in the identification of potential pharmacological treatments by studying the dynamic interactions between the most relevant proteins.

The use of GCNs in this study provides a unique and innovative approach to understanding the complex molecular mechanisms of COVID-19, offering valuable insights for future treatment and diagnostic tool development. However, this method may need further validation before becoming a routine diagnostic tool.

Overall, the results provided in this work contribute to gaining new insights into the mechanisms of the disease and how it affects the immune system and how nonlinear relationship between "message passing" proteins could particularly explain disease severity modulation during the Sars-CoV-2 infection.

**Table 2.  Variable descriptions of the clinical dataset.**

| Variable (subject_id) | Description (subject ID) |
|---|---|
| COVID | COVID status (tested positive prior to enrolment or during hospitalization) 0 = negative 1 = positive |
| Age cat | Age category 1 = 20–34 2 = 36–49 3 = 50–64 4 = 65–79 5 = 80+ |
| BMI cat | Body mass index: 0 = <18.5 (underweight) 1 = 18.5–24.9 (normal) 2 = 25.0–29.9 (overweight) 3 = 30.0–39.9 (obese) 4 = ≥40 (severely obese) 5 = Unknown |
| HEART | Pre-existing heart disease? HEART - (coronary artery disease, congestive heart failure, valvular disease) 0 = No 1 = Yes |
| LUNG | Pre-existing lung disease? LUNG - (asthma, COPD, requiring home O2, any chronic lung condition) 0 = No 1 = Yes |
| KIDNEY | Pre-existing kidney disease? KIDNEY - (chronic kidney disease, baseline creatinine >1.5, ESRD) 0 = No 1 = Yes |
| DIABETES | Pre-existing diabetes? DIABETES - (pre-diabetes, insulin and non-insulin dependent diabetes) 0 = No 1 = Yes |
| HTN | Pre-existing hypertension - HTN 0 = No 1 = Yes |
| IMMUNO | Pre-existing immunocompromised condition ? IMMUNO (active cancer, chemotherapy, transplant, immunosuppressant agents, aspenic) 0 = No 1 = Yes |
| Resp_Symp | Respiratory symptoms? Symp_Resp (sore throat, congestion, productive or dry cough, shortness of breath or hypoxia, or chest pain) 0 = No 1 = Yes |
| Fever_Sympt | Febrile symptom |
| GI_Symp | Any GI related symptoms at presentation (abdominal pain, nausea, vomiting, diarrhea) |
| WHO 0 | WHO score for day 0 study window - enrollment plus 24 h 1 = Death within 28 d 2 = Intubated, ventilated, survived to 28 d 3 = Non-invasive ventilation or high-flow nasal cannula 4 = Hospitalized, supplementary O2 required 5 = Hospitalized, no supplementary O2 required 6 = Not hospitalized |
| WHO 3 | WHO score for day 3 study window 1 = Death within 28 d 2 = Intubated, ventilated, survived to 28 d 3 = Non-invasive ventilation or high-flow nasal cannula 4 = Hospitalized, supplementary O2 required 5 = Hospitalized, no supplementary O2 required 6 = Not hospitalized |
| WHO 7 | WHO score for day 7 study window 1 = Death within 28 d 2 = Intubated, ventilated, survived to 28 d 3 = Non-invasive ventilation or high-flow nasal cannula 4 = Hospitalized, supplementary O2 required 5 = Hospitalized, no supplementary O2 required 6 = Not hospitalized |
| WHO 28 | WHO score on study day 28 1 = Death within 28 d 2 = Intubated, ventilated, survived to 28 d 3 = Non-invasive ventilation or high-flow nasal cannula 4 = Hospitalized, supplementary O2 required 5 = Hospitalized, no supplementary O2 required 6 = Not hospitalized |
| WHO max | WHO max category at 28 d (maximum WHO score within first 28 d with death being the maximum possible) 1 = Death within 28 d 2 = Intubated, ventilated, survived to 28 d 3 = Non-invasive ventilation or high-flow nasal cannula 4 = Hospitalized, supplementary O2 required 5 = Hospitalized, no supplementary O2 required 6 = Not hospitalized |
| abs_neut_0_cat | Absolute neutrophil count day 0 category: 1 = 0–0.99 2 = 1.0–3.99 3 = 4.0–7.99 4 = 8.0–11.99 5 = 12+ |
| abs_lymph_0_cat | Absolute lymphocyte count day 0 category: 1 = 0–0.49 2 = 0.50–0.99 3 = 1.00–1.49 4 = 1.50–1.99 5 = 2+ |
| abs_mono_0_cat | Absolute monocyte day 0 category 1 = 0–0.24 2 = 0.25–0.49 3 = 0.50–0.74 4 = 0.75–0.99 5 = 1.0+ |
| creat_0_cat | Creatinine day 0 category 1 = 0–0.79 2 = 0.80–1.19 3 = 1.20–1.79 4 = 1.80–2.99 5 = 3+ |
| crp_0_cat | c-reactive protein day 0 category: 1 = 0–19.9 2 = 20–59.0 3 = 60–99.9 4 = 100–179 5 = 180+ |
| ddimer_0_cat | D-dimer day 0 category: 1 = 0–499 2 = 500–999 3 = 1000–1999 4 = 2000–3999 5 = 4000+ |
| ldh_0_cat | Lactate dehydrogenase day 0 category: 1 = 0–200 2 = 200–299 3 = 300–399 4 = 400–499 5 = 500+ |
| Trop_72h | Cardiac event ? Trop_72h - (hs-cTn =>100 within first 72 h of presentation) 0 = No 1 = Yes |
| abs_neut_3_cat | Absolute neutrophil count day 3 category: 1 = 0–0.99 2 = 1.0–3.99 3 = 4.0–7.99 4 = 8.0–11.99 5 = 12+ |
| abs_lymph_3_cat | Absolute lymphocyte count day 3 category: 1 = 0–0.49 2 = 0.50–0.99 3 = 1.00–1.49 4 = 1.50–1.99 5 = 2+ |
| abs_mono_3_cat | Absolute monocyte count day 3 category: 1 = 0–0.24 2 = 0.25–0.49 3 = 0.50–0.74 4 = 0.75–0.99 5 = 1.0+ |
| creat_3_cat | Creatinine day 3 category 1 = 0–0.79 2 = 0.80–1.19 3 = 1.20–1.79 4 = 1.80–2.99 5 = 3+ |
| crp_3_cat | c-reactive protein day 3 category: 1 = 0–19.9 2 = 20–59.0 3 = 60–99.9 4 = 100–179 5 = 180+ |
| ddimer_3_cat | D-dimer day 3 category: 1 = 0–499 2 = 500–999 3 = 1000–1999 4 = 2000–3999 5 = 4000+ |
| ldh_3_cat | Lactate dehydrogenase day 3 category: 1 = 0–200 2 = 200–299 3 = 300–399 4 = 400–499 5 = 500+ |
| abs_neut_7_cat | Absolute neutrophil count day 7 category: 1 = 0–0.99 2 = 1.0–3.99 3 = 4.0–7.99 4 = 8.0–11.99 5 = 12+ |
| abs_lymph_7_cat | Absolute lymphocyte count day 7 category: 1 = 0–0.49 2 = 0.50–0.99 3 = 1.00–1.49 4 = 1.50–1.99 5 = 2+ |

**Table 2.  Continued**

| Variable (subject_id) | Description (subject ID) |
| --- | --- |
| abs_mono_7_cat | Absolute monocyte count day 7 category: 1 = 0–0.24 2 = 0.25–0.49 3 = 0.50–0.74 4 = 0.75–0.99 5 = 1.0+ |
| creat_7_cat | Creatinine day 7 category 1 = 0–0.79 2 = 0.80–1.19 3 = 1.20–1.79 4 = 1.80–2.99 5 = 3+ |
| crp_7_cat | c-reactive protein day 7 category: 1 = 0–19.9 2 = 20–59.0 3 = 60–99.9 4 = 100–179 5 = 180+ |
| ddimer_7_cat | D-dimer day 3 category: 1 = 0–499 2 = 500–999 3 = 1000–1999 4 = 2000–3999 5 = 4000+ |
| ldh_7_cat | Lactate dehydrogenase day 7 category: 1 = 0–200 2 = 200–299 3 = 300–399 4 = 400–499 5 = 500+ |

# Materials and Methods

### Samples

Data were provided by the Massachusetts General Hospital Emergency Department COVID-19 with Olink proteomic (publicly available at https://www.olink.com/mgh-covid-study/). Clinical dataset and plasma proteomes of 384 adult patients were distributed in 306 (80%) patients that tested positive in COVID-19 that were named as *Covid +* and 78 patients tagged as *Covid –* that tested negative, even suffering from respiratory symptoms. Blood samples were drawn from *Covid +* patients on days 0, 3, and 7 after their admissions. The blood samples from *Covid –* patients were only drawn on day 0. The clinical dataset reported the 28 d outcome stratification according with the WHO scores of maximal severity described. Additionally, other clinical parameters including comorbidities such as patient age or preexisting medical complications (i.e., chronic kidney disease, any chronic lung condition such as asthma, diabetes) were also recorded. To anonymise this dataset, continuous variables were turned into categorical (see Table 2 and Supplemental Datas 1 and Supplemental Datas 2).

### Interpreting latent space of clinical features

The Shapley values are a widely used approach in cooperative game theory which have advantageous properties, as described by Yoshida et al (2020). These properties facilitate a clear understanding of the computation and interpretation of Kapley values as explanations for machine learning models. This is achieved through an empirical approach using the shap Python package, which demonstrates the explanations for increasingly complex models (Lundberg & Lee, 2017). In this study, we apply Shapley values to explain the predictions made by a neural network model trained on clinical variables. The model was trained using the "adam" optimizer and "binary cross entropy" loss functions (Chollet et al, 2015) to uncover why it makes different predictions for different individuals. To use SHAP, the model must take a 2D numpy array as input; so, a wrapper function was defined around the original Keras predict function (Chollet et al, 2015). To explain a single prediction (as shown in Fig 3A), 50 samples from the dataset were selected to represent typical feature values and 500 perturbation samples were used to estimate the SHAP values, which required 500 * 50 evaluations of the model. To explain many predictions, the above process was repeated for 50 individuals. Note that as this explanation is based on a sampling approximation, each explanation may take a few seconds, depending on the machine setup (as shown in Fig 3B–D).

### Notes on persistent homology (multi-topology)

Persistent homology is a mathematical theory used as a tool in topological data analysis (TDA) to study the topological features of a dataset (i.e., plasma proteome of each patient in the cohort). It is a type of algebraic topology that uses algebraic invariants to study the shapes of data. The main idea behind persistent homology is to study how the topological features of a dataset change as the scale of observation changes. This is often done by constructing a sequence of simplicial complexes from the data, each of which represents the data at a different scale, and then studying how the topological features of these simplicial complexes evolve over time.

In TDA, persistent homology is used to identify topological features of a dataset that are persistent, or stable, over a range of scales (i.e., multi-scale topology). These persistent features are considered to be the most significant topological features of the data, and they can be used to distinguish different classes of data or to identify patterns in the data. To use persistent homology in TDA, one typically starts by constructing a simplicial complex from the data, and then applying algebraic topology techniques to study the topological features of the complex. This can be done using specialised software tools, such as GUDHI, DIPHA, or Dionysus (Maria et al, 2014; Gillani et al, 2016; Tauzin et al, 2020), which are designed to compute persistent homology and other topological invariants.

### Notes on the GCN

We endowed the regulatory persistent-based protein networks with a convolutional design (i.e., neural networks) along with a spectral rule of node aggregation as in GCN (Defferrard et al, 2016). The sequential combination of two hybrid models enabled the learning of interactions between ACE2 and TMPRSS2, and the persistent proteins needed by the virus to be spread in cells. Following this reasoning, we primarily made use of the identity matrix $I$ as features and the adjacency matrix $A$ (Model 1) contributing the model in the following spectral rule:

$$S_r(A, I)_i = \sum_{l=1}^{N} \frac{1}{D_{k,k}^{1/2}} A_{k,l} \frac{1}{D_{l,l}^{1/2}} I_l,$$

where $D$ is the degree matrix. Right after, we considered the metric given by distance of the shortest path to characterise the early

aggregation of persistent proteins to ACE2 and TMPRSS2 as an additional feature in a second model (Model 2). Thus, we generated models with two layers computed by 32 units per layer and a 2D transformation of the activation function tanh. When applying the spectral rule, the relu activation function is applied at the beginning of the layer implementation instead of later on. The number of epochs was set to 250 and 5,000, respectively. We computed the stochastic gradient descent optimizer in the training task picking a learning rate and momentum regularization set to 0.001 and flagged true, respectively. The semi-supervised classification (*covid* or *non–covid* early proteins aggregation) of nodes in the persistent-based protein networks (Kipf & Welling, 2017) was performed by an in-house python script based on MXNet implementation (Faster Cheaper Leaner, 2021).

### Ethics Statement

The Partners Human Research Committee approved the collection and analysis of samples and waived the requirement for informed consent.

## Data Availability

All data used in this work are included in the article and/or supplementary material.

## Supplementary Information

## Acknowledgements

We would like to thank the funding from the National Research Association (ANR) (Inflamex renewal 10-LABX-0017 to I Morilla), Consejería de Universidades, Ciencias y Desarrollo, fondos FEDER de la Junta de Andalucía (ProyExec_0499 to I Morilla), DHU FIRE Emergence 4, and the l'Agence de la Biomédecine.

### Author Contributions

S Gauthier: data curation and formal analysis.
A Tran-Dinh: methodology, and writing—original draft, review, and editing.
I Morilla: conceptualization, data curation, formal analysis, supervision, funding acquisition, investigation, visualization, methodology, project administration, and writing—original draft, review, and editing.

### Conflict of Interest Statement

The authors declare that they have no conflict of interest.

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
