## [Reviewer comments · Life Science Alliance]

Life Science Alliance

Plasma Proteome Predicts COVID-19 Severity with Graph Convolutional Network and Multi-Scale Topology

Samy Gauthier, Alexy Tran-Dinh, and Ian Morilla

DOI: <https://doi.org/10.26508/lsa.202201624>

Corresponding author(s): Ian Morilla, Sorbonne Paris Cité

Review Timeline:

Submission Date:	2022-07-22
Editorial Decision:	2022-10-05
Revision Received:	2022-12-13
Editorial Decision:	2023-01-13
Revision Received:	2023-02-06
Accepted:	2023-02-06

Scientific Editor: Novella Guidi

Transaction Report:

October 5, 2022

Re: Life Science Alliance manuscript #LSA-2022-01624-T

Dr. Ian Morilla
Sorbonne Paris Cité
Institut Galilée, Université Sorbonne Paris Nord
99, avenue Jean-Baptiste Clément - Villetaneuse
Villetaneuse, Paris 93430
France

Dear Dr. Morilla,

Thank you for submitting your manuscript entitled "Proteome dynamics of COVID-19 severity learnt by a graph convolutional network of multi-scale topology". The manuscript has been evaluated by expert reviewers, whose reports are appended below. Unfortunately, after an assessment of the reviewer feedback, our editorial decision is against publication in Life Science Alliance.

Although your manuscript is intriguing, I feel that the points raised by the reviewers are more substantial than can be addressed in a typical revision period. If you wish to expedite publication of the current data, it may be best to pursue publication at another journal.

Given the interest in the topic, I would be open to re-submission to Life Science Alliance of a significantly revised and extended manuscript that fully addresses the reviewers' concerns and is subject to further peer review. If you would like to resubmit this work to Life Science Alliance, you may submit an appeal directly through our manuscript submission system. Please note that priority and novelty would be reassessed at re-submission.

Regardless of how you choose to proceed, we hope that the comments below will prove constructive as your work progresses.

Thank you for thinking of Life Science Alliance as an appropriate place to publish your work.

Sincerely,

Reviewer #1 (Comments to the Authors (Required)):

In this manuscript, Gauthier et al. used mathematical models in order to evaluate plasma proteins and correlate those proteins with the severity of COVID-19. Although, it is a very interesting and important work I believe that there are some aspects that need to be addressed, namely:

- The text from line 47 to the end of the introduction, correspond in fact to results from the work - is not a real introduction to the theme-, thus my suggestion is to add a new subsection in the results devoted to the study design to include this text. Moreover, I also believe that the work will benefit from an initial scheme representing the overall analysis.
- The names of the supplementary files do not correspond to the information provided by the authors in the text, which make it difficult to establish the correct association. Moreover, considering the information available in the "variable description" file added as a supplementary file, it is my opinion that this information should be indicated in the main manuscript in the form of a table (for instance in the Methods section). In my opinion, it is important that the variables used are easily accessible to the readers. In a similar way, more information is needed regarding the proteomics data: is this data from a public repository? If yes, please indicate the code of the project; if not, the authors must indicate how the data was acquired and add the proteomics results to the manuscript.
- The order of appearance of the images should correspond to their reference in the text, for instance, Fig. S4 is explained prior to Fig. S3. In a similar way, the different elements that constitute a figure should be also organized in a more coherent way, for example, in figure 1 the authors present first the element D and only after that the element C (which makes sense, so please change the position of the elements); in figure 2 the authors start explaining the element D and only after that the element A, etc, etc.
- Still regarding the figures, please be sure that all the elements are indicated, such as de axis legends, the color code explanation, etc, etc. Some of this information is missing in some figures.

- The last section of the results "Dynamic tracking of protein interactions ..." presents lots of information regarding different sets of proteins, however, there is any visual element supporting this section which makes it very difficult to follow. In this sense, it is my opinion that this section - which is the most important section of the work by being focused on the identification of different protein patterns related to the disease severity - needs to be better supported with the presentation of the results in the form of figures or tables. In a similar way, it is not clear to me how the authors obtained the values of accuracy referred to in lines 178 to 182; where are these results?
- In my opinion, the methods section is too incomplete.
- Finally, the authors indicate that they obtained functional evidence on how particular sequences of proteins interact with the virus (lines 258 to 259), however, the author didn't perform any kind of experimental studies to prove their findings. In fact, what the authors have is a potential model, which needs to be confirmed with independent studies. In this sense, the authors need to clearly transmit the idea that what they obtained in this work is a hypothesis that needs validation. Moreover, the authors also need to justify the use of plasma instead of the use of immune cells, and considering that several of their findings point to the potential mechanism of escape of the virus from the immune response, the author should also comment on how they think that their finding in plasma correlate well with the real response of the immune system (since they do not evaluate the cells). Considering this, I also believe that the title should also reflect the fact that the work was performed in plasma, thus instead of the current title I think that the title should be "Plasma proteome dynamics of COVID-19 severity (...)".

Reviewer #2 (Comments to the Authors (Required)):

This manuscript, authored by Gauthier et al, described a study that modeled plasma proteome of 384 COVID-19 patients. The authors attempted to take into consideration of clinical evaluation outcome of each patient in order to identify severity of the disease using graph convolutional network.

While the authors are commendable for their approach, the group appears to be lack of experience in writing biomedical research articles. There is very little description about the clinical information about the 384 COVID patients other than supplemental tables with scores assigned. It is unclear how, where, and when the plasma samples were acquired and stored and analyzed at proteome level. Without this information, it is difficult to assess the rigor of the study.

The data analyses mainly used methods that are unfamiliar to biomedical researchers. Most importantly, the analyses are purely conceptual without any experimental data to back up. The authors also failed to interpret the results in a way that will guide experimental researchers to test new ideas. For example, Figure 4 describes gene regulatory networks involving ACE2 and TMPRSS2. These networks were stratified into mild, intermediate and acute patients. No definition was given to these groups and no meaningful interpretation was derived from this analysis. Are the authors intending to say these gene networks predict disease progression or are just associated with disease progression?

Given all these, this reviewer feels the manuscript is more suitable to a computational biology journal.

Reviewer #1 (Comments to the Authors (Required)):

In this manuscript, Gauthier et al. used mathematical models in order to evaluate plasma proteins and correlate those proteins with the severity of COVID-19. Although, it is a very interesting and important work I believe that there are some aspects that need to be addressed, namely:

- The text from line 47 to the end of the introduction, correspond in fact to results from the work - is not a real introduction to the theme-, thus my suggestion is to add a new subsection in the results devoted to the study design to include this text. Moreover, I also believe that the work will benefit from an initial scheme representing the overall analysis.

- **Reply: Thanks to the reviewer for this constructive suggestion. Now, we have moved those paragraphs to “Results” into a subsection named “Study design”, completing and rewording the whole “introduction” section. Regarding the initial scheme, we have included the Figure 1 describing the entire flowchart followed in this study.**

- The names of the supplementary files do not correspond to the information provided by the authors in the text, which make it difficult to establish the correct association. Moreover, considering the information available in the "variable description" file added as a supplementary file, it is my opinion that this information should be indicated in the main manuscript in the form of a table (for instance in the Methods section). In my opinion, it is important that the variables used are easily accessible to the readers. In a similar way, more information is needed regarding the proteomics data: is this data from a public repository? If yes, please indicate the code of the project; if not, the authors must indicate how the data was acquired and add the proteomics results to the manuscript.

- **Reply: Thanks for pointing this out and the later suggestions. Now, all the supplementary files correspond to the information provided in the manuscript. Additionally, we have added some tables on the “Methods” and “Results” section of the manuscript describing variables used in the analysis. And the same for details on the proteomics data, which are publicly available at <https://www.olink.com/mgh-covid-study/>, just as they are now indicated in the manuscript.**

- The order of appearance of the images should correspond to their reference in the text, for instance, Fig. S4 is explained prior to Fig. S3. In a similar way, the different elements that constitute a figure should be also organized in a more coherent way, for example, in figure 1 the authors present first the element D and only after that the element C (which makes sense, so please change the position of the elements); in figure 2 the authors start explaining the element D and only after that the element A, etc, etc.

- **Reply: We apologise for this mistake. We have rearranged figures and their descriptions in the manuscript accordingly.**

- Still regarding the figures, please be sure that all the elements are indicated, such as de axis legends, the color code explanation, etc, etc. Some of this information is missing in some figures.

- **Reply: Thanks for pointing this out! We carefully fixed this issue throughout the manuscript, including supplementary figures.**

- The last section of the results "Dynamic tracking of protein interactions ..." presents lots of information regarding different sets of proteins, however, there is any visual element supporting this section which makes it very difficult to follow. In this sense, it is my opinion that this section - which is the most important section of the work by being focused on the identification of different protein patterns related to the disease severity - needs to be better supported with the presentation of the results in the form of figures or tables. In a similar way, it is not clear to me how the authors obtained the values of accuracy referred to in lines 178 to 182; where are these results?

- **Reply: This is another constructive comment. Thanks to the reviewer for the suggestion. Unfortunately, results are dynamic systems learnt from data already displayed in figures introduced in the manuscript. By their intrinsic nature, it could mislead to display those results statically. We must address readers to view/watch the linked videos. However, we have included in "Results" section a table describing some of those results and the others have been added on the supplementary manuscript.**
- **Regarding results shown in lines 178 to 182, we have included some explanatory tables on the manuscript as well as new figures in the supplementary information that clarify the algorithm such results come from.**

- In my opinion, the methods section is too incomplete.

- **Reply: Excellent suggestion, thanks! To meet reviewer's comment, we have added some subsections on the "Methods" section completing so that part of the manuscript. Likewise, the supplementary information with new explanatory figures.**

- Finally, the authors indicate that they obtained functional evidence on how particular sequences of proteins interact with the virus (lines 258 to 259), however, the author didn't perform any kind of experimental studies to prove their findings. In fact, what the authors have is a potential model, which needs to be confirmed with independent studies. In this sense, the authors need to clearly transmit the idea that what they obtained in this work is a hypothesis that needs validation. Moreover, the authors also need to justify the use of plasma instead of the use of immune cells, and considering that several of their findings point to the potential mechanism of escape of the virus from the immune response, the author should also comment on how they think that their finding in plasma correlate well with the real response of the immune system (since they do not evaluate the cells). Considering this, I also believe that the title should also reflect the fact that the work was performed in plasma, thus instead of the current title I think that the title should be "Plasma proteome dynamics of COVID-19 severity (...)".

>**Reply: Thanks for these useful suggestions and comments. Unfortunately, experimental validation on COVID inpatients is beyond the scope of this study since at present our group is only computational. We may want to mention our group is now collaborating with some clinicians at Bichat hospital in Paris to experimentally validate our model. In any case, we have reworded "Discussion" section to make clear our results derived from a model that possibly would need further validation. Regarding the**

title specifying the use of plasma protein, we agree with the reviewer and included so the word “plasma” before proteome in the main title of the manuscript.

Reviewer #2 (Comments to the Authors (Required)):

This manuscript, authored by Gauthier et al, described a study that modeled plasma proteome of 384 COVID-19 patients. The authors attempted to take into consideration of clinical evaluation outcome of each patient in order to identify severity of the disease using graph convolutional network.

While the authors are commendable for their approach, the group appears to be lack of experience in writing biomedical research articles. There is very little description about the clinical information about the 384 COVID patients other than supplemental tables with scores assigned. It is unclear how, where, and when the plasma samples were acquired and stored and analyzed at proteome level. Without this information, it is difficult to assess the rigor of the study.

- **Thanks, that is an excellent comment. We have now reworded and completed the “introduction” and “Methods” sections introducing details on dataset/cohort and the protocol followed to measure proteome levels. Additionally, we have included the Figure 1 describing the entire flowchart followed in this study.**

The data analyses mainly used methods that are unfamiliar to biomedical researchers. Most importantly, the analyses are purely conceptual without any experimental data to back up.

>Reply: Thanks for these useful suggestions and comments. Unfortunately, experimental validation on COVID inpatients is beyond the scope of this study since at present our group is only computational. We may want to mention our group is now collaborating with some clinicians at Bichat hospital in Paris to experimentally validate our model. In any case, we have reworded “Discussion” section to make clear our results derived from a model that possibly would need further validation.

The authors also failed to interpret the results in a way that will guide experimental researchers to test new ideas. For example, Figure 4 describes gene regulatory networks involving ACE2 and TMPRSS2. These networks were stratified into mild, intermediate and acute patients. No definition was given to these groups and no meaningful interpretation was derived from this analysis. Are the authors intending to say these gene networks predict disease progression or are just associated with disease progression?

- **Reply: Thanks, we apologise for not making that point clear enough in the manuscript. We have reworded accordingly and now the reader could find the following explanatory paragraphs just as:**
- **Lines 90-116: “In order to better understand and track disease progression, we propose using information theory scores, as proposed by Shannon ...
... In particular, we obtain a discrete random variable V ...
... The generalised continuous normal random variable yielded the best performance...**

**... we can accurately track disease progression...
... By means of this function, we calibrated hierarchical clustering models that were computed applying the Hdbscan algorithm...
... Hence, we achieved to discriminate the Covid cohort into three different groups...
... Those groups basically met the mild, intermediate, and acute symptoms as registered in the available clinical dataset (see Fig.2d).”**

We have also added explanatory sentences in the "Methods" section, the "Results" section, and the caption for Figure 4 to clarify the predictive nature of the graph described in these sections. These sentences provide further information on how the graph was constructed and how it can be used to make predictions about the progression of COVID-19. This additional information is important for understanding the full significance of the graph and its implications for disease management and treatment.

Given all these, this reviewer feels the manuscript is more suitable to a computational biology journal.

- **Reply: Thanks for pointing this out! We might agree with reviewer, but our manuscript was transferred to *Life Science Alliance* upon Genome Research editor's suggestion who accounted for the globality/multidisciplinary of the former.**

January 13, 2023

RE: Life Science Alliance Manuscript #LSA-2022-01624-TR-A

Dr. Ian Morilla
Sorbonne Paris Cité
Institut Galilée, Université Sorbonne Paris Nord
99
Avenue Jean-Baptiste Clément
Villetaneuse, Paris 93430
France

Dear Dr. Morilla,

Thank you for submitting your revised manuscript entitled "Graph convolutional network learns plasma proteome dynamics that predict COVID-19 severity". We would be happy to publish your paper in Life Science Alliance pending final revisions necessary to meet our formatting guidelines.

- please address the remaining Reviewer 1's concerns
- please upload your manuscript text as an editable doc file
- please upload both your main and supplementary figures as single files
- please use the [10 author names, et al.] format in your references (i.e. limit the author names to the first 10)
- please add a figure legend section to your manuscript
- please rename your supplementary figures as Figure S1, Figure S2, etc.
- please provide an ethics statement before acknowledgment

A. FINAL FILES:

B. MANUSCRIPT ORGANIZATION AND FORMATTING:

Sincerely,

Reviewer #1 (Comments to the Authors (Required)):

I would like to thank the authors for the effort made to address the reviewers' comments/suggestions. In general, the authors addressed my comments, but there are some aspects that need to be better discussed prior to publication. In the abstract, the author state as a conclusion that: "The extracted topological invariants, which describe protein expression at different times, are used as the basis for a graph convolutional network. This model enables the dynamic learning of the molecular interactions between the identified proteins.". In my opinion, a more applied conclusion/explanation should be added. For instance, to what extent this model can be used to determine the gravity of an infection in an early moment, and thus provide a more adequate treatment to those patients, or how this model can be used to identify new potential pharmacological treatments that prevent the dynamic interactions/alterations observed. The introduction starts as follows "The blitzkrieg triggered". Please change the term blitzkrieg for something more scientific. This is mainly a war term, not frequent in scientific papers. Please re-write the new paragraph from lines 55 to 61 to be clearer and add references. The Discussion section also needs to be supported by references. There is no single reference in the entire section. Please change the name of the figures and tables from the supplementary file to Figure SX and table SX, to be in accordance with their reference in the main text. In the legends of some of the tables de authors indicate the following "First-appearing figures stand for model 1 while the second ones stand for model 2 with aggregation (see Methods).", it is not clear which figures they are referring to, as well as there is no clear explanation in the methods section regarding which is the model 1 and 2. I still think that the methods section does not fulfill the main goal of this section, which is having the information needed to repeat the work by others. Finally, I also miss some justification for why the approaches presented in this work are better than other more conventional methods and whether this can be easily implemented in/translated into the diagnostic routine. I think is important to justify the inclusion of this work mainly focused on computational methods in a generic or biological journal.

Reviewer #2 (Comments to the Authors (Required)):

I do not have additional comments.

Reviewer #1 (Comments to the Authors (Required)):

I would like to thank the authors for the effort made to address the reviewers' comments/suggestions. In general, the authors addressed my comments, but there are some aspects that need to be better discussed prior to publication.

- In the abstract, the author state as a conclusion that: "The extracted topological invariants, which describe protein expression at different times, are used as the basis for a graph convolutional network. This model enables the dynamic learning of the molecular interactions between the identified proteins.". In my opinion, a more applied conclusion/explanation should be added. For instance, to what extent this model can be used to determine the gravity of an infection in an early moment, and thus provide a more adequate treatment to those patients, or how this model can be used to identify new potential pharmacological treatments that prevent the dynamic interactions/alterations observed.

>> Thanks to the reviewer for this constructive suggestion! Now a new paragraph has been added in the abstract according to reviewer's comment.

- The introduction starts as follows "The blitzkrieg triggered". Please change the term blitzkrieg for something more scientific. This is mainly a war term, not frequent in scientific papers.

>> Thanks for pointing this out! We have replaced that term by "The sudden spread of".

- Please re-write the new paragraph from lines 55 to 61 to be clearer and add references. The Discussion section also needs to be supported by references. There is no single reference in the entire section.

>> This is another constructive comment. We have reworded and improved the understanding of those paragraphs. Now they should be clearer to readers.

- Please change the name of the figures and tables from the supplementary file to Figure SX and table SX, to be in accordance with their reference in the main text.

>> Thanks! Now, the entire set of figures and tables are arranged in accordance with their reference in the main text.

- In the legends of some of the tables de authors indicate the following "First-appearing figures stand for model 1 while the second ones stand for model 2 with aggregation (see Methods).", it is not clear which figures they are referring to, as well as there is no clear explanation in the methods section regarding which is the model 1 and 2.

>> We apologise for this mistake. A much clearer description of models 1 and 2 within the hybrid one has been included. Additionally, we have reworded the caption of that figure.

- I still think that the methods section does not fulfill the main goal of this section, which is having the information needed to repeat the work by others.

>> We apologise for this misunderstanding! We have addressed that reviewers' suggestion in the manuscript by adding some more explanatory sections.

- Finally, I also miss some justification for why the approaches presented in this work are better than other more conventional methods and whether this can be easily implemented in/translated into the diagnostic routine. I think is important to justify the inclusion of this work mainly focused on computational methods in a generic or biological journal.

>> We totally agree the reviewers' opinion. We have added a few more supporting paragraphs on the computational choice in the discussion section.

Reviewer #2 (Comments to the Authors (Required)):

I do not have additional comments.

>> We would like to thank reviewers' efforts in improving the final manuscript.

February 6, 2023

RE: Life Science Alliance Manuscript #LSA-2022-01624-TRR

Dr. Ian Morilla
Sorbonne Paris Cité
LAGA, Institut Galilée
99
Avenue Jean-Baptiste Clément
Villetaneuse, Paris 93430
France

Dear Dr. Morilla,

Thank you for submitting your Methods entitled "Plasma Proteome Predicts COVID-19 Severity with Graph Convolutional Network and Multi-Scale Topology". It is a pleasure to let you know that your manuscript is now accepted for publication in Life Science Alliance. Congratulations on this interesting work.

DISTRIBUTION OF MATERIALS:

Again, congratulations on a very nice paper. I hope you found the review process to be constructive and are pleased with how the manuscript was handled editorially. We look forward to future exciting submissions from your lab.

Sincerely,
